# Natural Zeolites for the Sorption of Ammonium: Breakthrough Curve Evaluation and Modeling

**DOI:** 10.3390/molecules28041614

**Published:** 2023-02-07

**Authors:** Stephan Eberle, Viktor Schmalz, Hilmar Börnick, Stefan Stolte

**Affiliations:** Institute of Water Chemistry, Technische Universität Dresden, 01062 Dresden, Germany

**Keywords:** breakthrough curve modeling, linear driving force model, Thomas model, two-factor variance analysis, ultrapure water and natural water matrices

## Abstract

The excessive use of ammonium fertilizer and its associated leakage threatens aquatic environments around the world. With a focus on the treatment of drinking water, the scope of this study was to evaluate and model the breakthrough curves for NH_4_^+^ in zeolite-filled, fixed-bed columns. Breakthrough experiments were performed in single- and multi-sorbate systems with the initial K^+^ and NH_4_^+^ concentrations set to 0.7 mmol/L. Breakthrough curves were successfully modeled by applying the linear driving force (LDF) and Thomas models. Batch experiments revealed that a good description of NH_4_^+^ sorption was provided by the Freundlich sorption model (R^2^ = 0.99), while unfavorable sorption was determined for K^+^ (n_F_ = 2.19). Intraparticle diffusion was identified as the rate limiting step for NH_4_^+^ and K^+^ during breakthrough. Compared to ultrapure water, the use of tap, river, and groundwater matrices decreased the treated bed volumes by between 25% and 69%—as measured at a NH_4_^+^ breakthrough level of 50%. The concentrations of K^+^ and of dissolved organic carbon (DOC) were identified as the main parameters that determine NH_4_^+^ sorption in zeolite-filled, fixed-bed columns. Based on our results, the LDF and Thomas models are promising tools to predict the breakthrough curves of NH_4_^+^ in zeolite-filled, fixed-bed columns.

## 1. Introduction

To meet worldwide food demands, the use of nitrogen (N) fertilizer in agricultural applications has increased by 985% between 1961 and 2020 [1]. The use of N fertilizer (mainly in the form of ammonium—NH_4_^+^) has several advantages: increased crop qualities; increased yields; and increased resistance to environmental stresses (e.g., limited water availability) [2,3].

Farmers in threshold and developing countries often use excessive amounts of N fertilizer because of difficulties in predicting the requirements of each specific plant and soil condition. This leads to N leakage of between 40% and 50% on average globally and has major impacts on both soil and water environments (e.g., increased oxygen demand and simulation of eutrophication) [4,5]. Under anaerobic groundwater conditions, NH_4_^+^ concentrations as high as 390 mg/L have been recorded in a coastal aquifer-aquitard system in the Pearl River Delta, China [6]. Considering that global food demands continue to increase, it is reasonable to assume that NH_4_^+^ levels will also increase across various water matrices and environmental niches.

Compliance with local threshold values is the determining factor when selecting a suitable process for the removal of NH_4_^+^ from drinking water. NH_4_^+^ is only considered harmful to humans at levels above 100 mg/kg body weight per day [7]. During drinking water treatment processes, the formation of toxicologically relevant NO_2_^−^ species (via nitrification) and harmful chlorination side products must also be considered. In addition, NO_3_^−^ can be reduced to NO_2_^−^ in the stomach, which causes methemoglobinemia in infants. Finally, NO_3_^−^ is a precursor of a variety of carcinogenic and teratogenic *N*-nitroso compounds [8].

To overcome these threats to human health, NH_4_^+^ contaminated water can be treated using a range of technologies and processes. Such processes include: biological treatment processes (activated sludge and biofilm); air-stripping; membrane processes; and ion exchange [9,10]. When considering the treatment processes listed above, high energy expenditures can be avoided by excluding aeration processes. The use of additional chemicals and high pressures can be bypassed by using natural zeolites as sorbents.

Natural zeolites, particularly clinoptilolite, are promising sorbents for use in the processing of drinking water. Natural zeolites are abundantly available and cheap [11,12,13]; they can be subsequently used as a sustainable, slow-release fertilizer in agriculture [14]; they can be regenerated in a cost-effective manner through a simple brine treatment [15,16,17]; they have a broad range of applications; and most importantly, they demonstrate outstanding NH_4_^+^ removal efficiencies [18].

Natural zeolites consist of three-dimensional AlO_4_^−^- and SiO_4_^−^-tetrahedrons covalently connected through oxygen atoms, which feature a negative surface charge. This surface charge can be selectively compensated by NH_4_^+^, which exchanges with Na^+^, K^+^, Ca^2+^, and Mg^2+^ ions [19]. 

Previous studies have thoroughly investigated the use of natural zeolites with respect to the removal of heavy metals [20,21,22], organic contaminants [23], and high NH_4_^+^ loads during wastewater treatment [23]. It has been reported that K^+^ is the main competitor to NH_4_^+^ due to their similar hydrated ionic radii and charge densities [24]. 

Recently, we investigated granular natural zeolites in fixed-bed columns for the removal of NH_4_^+^ from drinking water sources [18,25]. Here, we investigate the NH_4_^+^ breakthrough curves of finer zeolite particles in order to reduce the dimensions and run time of the zeolite columns.

Modeling of the NH_4_^+^ breakthrough curves promises to further reduce time-consuming laboratory investigations. To date, different models have been used to calculate breakthrough curves. These models include: the LDF model, the Thomas model, the Yoon–Nelson model and the dose response model. With the exception of the LDF model, all of these models were previously applied to the modeling of NH_4_^+^ breakthrough curves in zeolite-filled, fixed-bed columns. These previous studies investigated the influence of model water and wastewater matrices and, in doing so, enabled the extrapolation of zeolite sorption performance for a variety of NH_4_^+^ removal scenarios [26,27]. The weakness of these models is that the description of the sorption mechanisms is reduced to simplified mathematical relationships. The LDF model also makes simplified assumptions, but it also includes a detailed evaluation of equilibrium and kinetic behavior. To the best of our knowledge, applying the LDF model for the modeling of NH_4_^+^ breakthrough curves in zeolite-filled, fixed-bed columns has not previously been reported in the literature. To date, the LDF model is best known for modeling the sorption performance of activated carbon [28].

The main objective of this study was to investigate the influence of single- and multi-sorbate systems and of natural water matrices on the NH_4_^+^ breakthrough curves in natural zeolite-filled, fixed-bed columns. In a departure from previous studies, the LDF and Thomas models were used to model the breakthrough curves for NH_4_^+^ and to investigate the influence of its main competitor—K^+^. To achieve this, batch experiments and kinetic studies were conducted to determine the parameters necessary to model breakthrough curves. Furthermore, the influence of various water matrices (single- and multi-sorbate systems in ultrapure water, tap-, surface-, river- and groundwater) on the breakthrough curve of NH_4_^+^ was investigated. Here, we demonstrate a promising tool with which to predict the NH_4_^+^ breakthrough curves and to improve drinking water quality, using cost-effective techniques. Improving the quality of drinking water is increasingly relevant, especially in threshold and developing countries.

## 2. Results and Discussion

### 2.1. Modeling and Experimental Validation

Initially, the parameters of the Freundlich sorption model and for intraparticle diffusion were determined (see Table 1). The experimental isotherm and kinetic data can be found in Appendix A.

By comparing the correlation coefficients (R^2^, Table 1) for each water matrix, it can be seen that NH_4_^+^ sorption is well described by the Freundlich sorption model. Several studies propose the Langmuir sorption model for a better correlation [15,29,30,31], while others describe the Freundlich sorption model as more convenient [14,23,32,33]. As Lin et al. and Chen et al. described, it is most likely that ion exchange is the predominant sorption mechanism at NH_4_^+^ concentrations below 1000 mgN/L for clinoptilolite-like zeolites [29,34]. In a previous study, we added support to this assumption by balancing the initial and end point ion equivalents for the same NH_4_^+^ concentration (0.7 mmol/L) used in this study [18]. The Langmuir sorption model was not investigated here because the LDF model is based on Freundlich parameters. 

It is notable that the Freundlich parameters, K_F_ and n_F,_ differ greatly between the isotherms investigated. Comparatively low K_F_ values indicate a weak sorption of the sorbate to the sorbent, which is the case for K^+^ (sorbent loading is low at low concentrations). Jaskūnas et al. determined similar values for K_F_ (1.73 × 10^−5^ $L^{0.23}{*\mathrm{mg}}^{0.77}$/g) and n_F_ (4.28) by investigating the sorption of K^+^ to a natural zeolite (clinoptilolite) [35]. The low ability of K^+^ to exchange charge balancing cations in the natural zeolite’s framework can be related to the following characteristics: (1) after Ca^2+^, K^+^ represents the second highest concentration (Table 2); and (2) K^+^ is able to exchange Ca^2+^, Mg^2+^ and Na^+^, but due to its comparatively large hydrated radii, Ca^2+^ (highest concentration in the zeolite’s framework) and Mg^2+^ cannot be easily exchanged. In contrast, NH_4_^+^ is able to exchange all of the charge-balancing cations (Ca^2+^, Mg^2+^, Na^+^, K^+^) in the zeolite’s framework, which also results in higher K_F_ values.

Comparing the NH_4_^+^ isotherm without the influence of competing K^+^ ions (NH_4_^+^: K_F_ = 1.02 $L^{1.02}{*\mathrm{mg}}^{0.07}$/g) with the isotherm where K^+^ is present (NH_4_^+^, (+K^+^): K_F_ = 1.21 $L^{1.21}{*\mathrm{mg}}^{0.37}$/g) suggests that the presence of K^+^ strengthens the sorption of NH_4_^+^. The n_F_ values (n_F_(NH_4_^+^) = 0.93 > n_F_(NH_4_^+^, (+K^+^)) = 0.63) imply that a faster breakthrough of NH_4_^+^ occurs in the presence of competing K^+^ ions—see Figure 1a,b.

The isotherm for NH_4_^+^ in the absence of K^+^ displayed almost linear behavior and was reflected in the very similar values for K_F_ and n_F_. These results support the findings of our previous study, using the same zeolite type (8–16 mm grain size) and over a range of concentrations we deemed relevant to drinking water (c_0_(NH_4_^+^) = 1–110 mgN/L), a linear isotherm shape was observed [18].

In the next step, intraparticle diffusion kinetics were investigated in a differential circuit batch reactor. The impact of film diffusion was negated by operating at a filter velocity of 29 m/h. Without the influence of the competing K^+^, the fastest intraparticle diffusion rate was determined for NH_4_^+^ (k_s_a_v_ = 1.1 × 10^−4^ 1/s). As the intraparticle diffusion is dependent on the molecular weight of the sorbate and the temperature, it is not surprising that the diffusion of K^+^ is comparatively slower (k_s_a_v_ = 3.6 × 10^−5^ 1/s). This can be attributed to the (>2 x) greater molecular weight of K^+^ (39 g/mol) compared to NH_4_^+^ (18 g/mol). The molecular weight of K^+^ results in a hindered K^+^ transport—as a result of confinement and interaction of K^+^ with the pore. The use of a constant temperature in each experiment meant that this parameter did not influence intraparticle diffusion. The slowest intraparticle diffusion (k_s_a_v_ = 9.8 × 10^−5^ 1/s) was determined for NH_4_^+^ in the presence of K^+^. It is assumed that K^+^ not only competes for the same sorption sites, but also interacts with and reduces the rate of intraparticle diffusion for NH_4_^+^. Furthermore, a dependence on sorbate concentration or sorbent loading may occur as the result of a combined mechanism or as the result of a dependence of loading on the surface diffusion coefficient (D_S_) [28].

It is also not surprising that the mass transfer coefficients k_F_a_v_ for the film diffusion are much faster than those of the intraparticle diffusion (k_s_a_v_). The difference in k_F_a_v_ between NH_4_^+^ and K^+^ is related to their diffusion coefficients in the aqueous phase (D_L_). As the molecular weight of K^+^ is higher than that of NH_4_^+^, D_L_ is comparatively lower for K^+^, and this results in a reduced rate of film diffusion.

After determining all necessary parameters, the experimental data were compared to the LDF and Thomas models. All of the parameters displayed in Table 1 were implemented in the program LDF 2.6© to obtain the modeled breakthrough curves for NH_4_^+^ and K^+^. The nonlinear Thomas model curve fitting was performed using the program OriginPro 2023© (see also Appendix A). Figure 1a,b show comparisons of the experimental data with curves predicted by the LDF and Thomas models, respectively.

After 80 h, each breakthrough curve showed a complete saturation of the zeolites used. The breakthrough curve for NH_4_^+^ was represented by a typical S-shape both with and without the influence of K^+^. In contrast, K^+^ achieved a breakthrough level of 23% after only 30 min. The offset of approximately 23% and the much faster breakthrough of K^+^ can be related to its weak sorption onto zeolites (K_F_ = 8.39 × 10^−3^ $L^{2.19}{*\mathrm{mg}}^{-1.19}$/g), high abundance in the zeolite’s framework, partial washout from the zeolite used, and its unfavorable sorption onto the natural zeolites used (n_F_ > 1). In general, K^+^ concentrations up to 0.7 mmol/L (equivalent to 27 mg/L) are quite unusual in drinking water applications—typically K^+^ concentrations are found to be between 0.02 mmol/L and 0.1 mmol/L [18]. A relatively high K^+^ concentration was chosen to simulate a worst case scenario and to investigate the influence of K^+^ and NH_4_^+^ at the same concentrations (i.e., in a 1:1 ratio). It was assumed that the NH_4_^+^ breakthrough curve obtained in the presence of concentrations of K^+^ that were relevant to drinking water would be very similar to the NH_4_^+^ breakthrough curve obtained in the absence of K^+^.

A steady saturation of the zeolite-filled, fixed-bed column was observed in all cases, particularly pronounced for NH_4_^+^ in the absence of K^+^. The relatively delayed breakthrough in all cases indicate that intraparticle diffusion is the diffusion limited step. This outcome is also in accordance with the findings of Damodara Kannan and Parameswaran, who investigated the NH_4_^+^ uptake by clinoptilolite from swine wastewater permeate [26].

Even though the intraparticle diffusion rate was 1.12 times higher for NH_4_^+^ (without K^+^), and the sorption was more favorable for NH_4_^+^ in the presence of K^+^ (n_F_ = 0.63), the breakthrough of NH_4_^+^ in the absence of K^+^ was reduced. This is strong evidence for the competing effect of K^+^ on the sorption of NH_4_^+^ onto zeolites. Because NH_4_^+^ and K^+^ compete for many of the same sorption sites, a much faster NH_4_^+^ breakthrough occurred in the presence of K^+^.

With regards to the breakthrough curves for NH_4_^+^, both the LDF and Thomas models were successfully validated—the modeled breakthrough curves of NH_4_^+^ were in close agreement with the experimentally obtained breakthrough curves. For K^+^, the modeling was not as accurate. The LDF model did not adequately describe the breakthrough curve for K^+^ during the first 16 h, whereas the Thomas model was inadequate to describe behavior after 24 h. Furthermore, it was not possible to maintain a typical S-shape breakthrough curve for K^+^ using the LDF model. The nonlinear Thomas model fitting provided a much better S-shape breakthrough curve for K^+^. To obtain a better fitting of the Thomas model, mainly in the first 6 h, the empty bed contact time (EBCT) should be increased in future studies—2–5 min EBCT are recommended for drinking water treatment applications [28].

To the best of our knowledge, this study is the first report of the LDF model being applied to NH_4_^+^ breakthrough in the presence of K^+^ in zeolite-filled, fixed-bed columns. The LDF model described the experimental isothermal and kinetic data sufficiently well. Furthermore, the experimental data were successfully fitted using the nonlinear Thomas model.

From a practical point of view, time-consuming and extensive field investigations could be reduced by switching to time-saving laboratory experiments. In turn, these laboratory experiments, which are still time-consuming, can be made easier still—by using simple breakthrough investigations through application of the Thomas model. Thus, accurate predictions of the dimensions of decentralized zeolite filters and their breakthrough characteristics are highly promising areas.

### 2.2. Ammonium Breakthrough in Natural Water Matrices

In addition to modeling NH_4_^+^ breakthrough in zeolite-filled, fixed-bed columns in single- and multi-sorbate systems, the influence of natural water matrices on NH_4_^+^ breakthrough characteristics is also of great interest. Therefore, the effects of different alkali metals, earth alkali metals and dissolved organic carbon (DOC) were investigated, at concentrations typically found in natural water matrices, in order to evaluate their influence on NH_4_^+^ breakthrough. Here, tap water, a 1:10 mixture of Elbe river water and tap water, Elbe river water, groundwater, and ultrapure water as a control were used as matrices.Table 3 and Appendix A summarize the initial cation and DOC concentrations as well as providing general information on the different matrices investigated.

The cation and DOC concentrations shown in Table 3 represent typical matrix-dependent concentrations. In a previous study, the sorption of NH_4_^+^ onto the same natural zeolite was most affected by the presence of K^+^, followed by Na^+^, Mg^2+^, and Ca^2+^ [18]. This is generally in agreement with other reported studies (although these can differ with respect to Mg^2+^ and Ca^2+^ [36,37]). Furthermore, zeolites can be used to remove DOC in a targeted manner [33] and in more complex applications [38]. It is suspected that macromolecular humic compounds function to block zeolite pores, or function as an additional molecular sieve (providing diffusion resistance) for NH_4_^+^, resulting in minor equilibrium loadings in batch investigations [18,28]. This is why the parameters that influence NH_4_^+^ sorption onto natural zeolites, mainly K^+^ and DOC concentrations, are of particular interest. The elevated cation concentrations found in groundwater and the higher DOC concentration found in the Elbe river water (see Table 3) can influence the breakthrough of NH_4_^+^. The effects of different, natural water matrices on NH_4_^+^ breakthrough curves are shown in Figure 2.

Compared to the control with ultrapure water, an accelerated NH_4_^+^ breakthrough was observed for all natural water matrices. Tap water and Elbe-/tap water exhibit a very similar breakthrough behavior. As the treated bed volume (BV) increases further, the Elbe-/tap water mixture exhibits a higher rate of breakthrough than tap water. The breakthrough of groundwater and Elbe river water is almost the same over the range of treated BV. To achieve a breakthrough level of 50%, the following matrix-dependent BVs were treated: 435 BV for ultrapure water, 325 BV for tap water, 268 BV for the Elbe-/tap water mixture, 176 BV for groundwater, and 135 BV for Elbe river water. Compared to the control, it is remarkable that the BV that was treated before a breakthrough of 50% was achieved decreased by 25% for tap water, by 38% for the Elbe-/tap water mixture, by 59% for groundwater and by 69% for Elbe river water. The typical S-shapes, observed for the breakthrough curves in single- and multi-sorbate systems (Section 2.1), could barely be seen for natural water matrices and supported the idea of a faster NH_4_^+^ breakthrough. An increase in the EBCT, from 2.16 min up to 5 min, should be employed in future studies and especially in field applications.

It is most likely that the concentrations of K^+^ and DOC were the determining factors in the shift of the breakthrough curves to the left (breakthrough occurring at earlier time points and lower treated BVs in the presence of these materials). The competing effect of K^+^ on NH_4_^+^ breakthrough can be interpreted by considering the physical characteristics of both ions as well as the dimensions of the framework of natural zeolites—shown in Table 4 and Table 5.

Zeolites are three-dimensional solids with well-defined structures. Natural zeolites are built around three ring types, each with different channel dimensions. Different ring types provide different sorption sites for hydrated cations. Ca^2+^ and Mg^2+^ possess relatively large hydrated radii compared to the other cations investigated. Due to the larger hydrated radii, as well as the molecular sieving effects that occur within natural zeolites, Ca^2+^ and Mg^2+^ cannot diffuse to all possible sorption sites (only to the center of horizontal ten-membered rings and horizontal eight-membered rings). Furthermore, it is thought that electrostatic repulsion of Ca^2+^ and Mg^2+^ can occur due to their relatively high charge densities. Similarly to Ca^2+^ and Mg^2+^, Na^+^ can only occupy certain sorption sites [40]. In contrast, NH_4_^+^ and K^+^, with identical hydrated radii and charge densities, can diffuse into and be exchanged within all ring types of natural zeolites [40]. Therefore, K^+^ is the main competitor to NH_4_^+^ and the most important cation to consider when discussing NH_4_^+^ breakthrough in zeolite-filled, fixed-bed columns. 

The influence of different anions, those that typically occur in natural water matrices, can be excluded. Even at high anion concentrations, far outside what is usually found in drinking water (3.5 mmol/L for NO_2_^−^, NO_3_^−^, HPO_4_^2−^/H_2_PO_4_^−^ and SO_4_^2−^), no significant effect on the loading of NH_4_^+^ onto the same natural zeolite was observed in batch experiments [18]. 

The point of zero charge (pH_PZC_) of the same natural zeolite was observed to be in the range between pH_PZC_ 6.24 and 6.47 [18], indicating that the surface is mostly positively charged at pH < pH_PZC_. Since the pH values used in our natural water matrices were higher than the pH_PZC_, it can be assumed that the zeolites’ surfaces are mostly negatively charged. Thus, the surface will repel the negative zeta potential of humic compounds in the DOC at pHs > 1.6. Nevertheless, it is suspected that the comparatively high DOC concentration in the Elbe river water (due to macromolecular humic compounds) functioned to block pores, or as an additional molecular sieve for NH_4_^+^, resulting in faster NH_4_^+^ breakthrough [18,28].

#### Influence of Potassium and the DOC on the Ammonium Breakthrough Curve

DOC and K^+^ concentrations in natural water matrices are thought to increase the rate of NH_4_^+^ breakthrough in zeolite-filled, fixed-bed columns. Figure 3a,b shows how DOC and K^+^ influence the Thomas model rate constant k_Th_, respectively. All Thomas model constant determined and modeled NH_4_^+^ breakthrough curves can be found in Appendix A and Appendix A.

The dependency of k_Th_ on the DOC (R^2^ = 0.98) and K^+^ (R^2^ = 0.94) concentration in natural water matrices is well described by a linear fit. According to Figure 3, a higher DOC and K^+^ concentration results in a higher k_Th_ and an accelerated breakthrough of NH_4_^+^. So, in addition to K^+^ (see Section 2.2), the DOC concentration also has a strong influence on the breakthrough of NH_4_^+^ in zeolite-filled, fixed-bed columns.

In order to assess whether K^+^ or the DOC has a stronger influence on k_Th_, a two-factor variance analysis was carried out by using the program OriginPro 2023© in Figure 4.

Figure 4 shows that the DOC has a stronger influence on k_Th_ than potassium, especially at concentrations > 6 mg/L. There was a significant dependency between k_Th_ and the DOC concentration, which is an important finding for field applications.

To further confirm the dependency between k_Th_ and K^+^ and the DOC, more natural water matrices should be investigated.

Nevertheless, due to the complex composition of DOC, it is challenging to describe a precise mechanism for the influence of DOC on k_Th_ and the breakthrough of NH_4_^+^ in zeolite-filled, fixed-bed columns. DOC consists of about 50% humic compounds, with relatively large molecular structures (humic and fulvic acids). The remainder is made up of low-molecular acidic and neutral substances and other compounds [28]. Because of the variety of molecular structures and weights, no individual species can be identified and analytically determined.

In summary, breakthrough curves of NH_4_^+^ in zeolite-filled, fixed-bed columns can be better predicted, and location-dependent estimations of k_Th_ can be made by: (1) measuring K^+^ and DOC concentrations; (2) applying the linear equations of Figure 3; and (3) applying a two-factor variance analysis.

## 3. Materials and Methods

### 3.1. Zeolite Characteristics

CLP85+, the natural zeolite (clinoptilolite) under investigation, was supplied by Zeolith Umwelttechnik Berlin GmbH, Berlin, Germany. In Table 2, the chemical composition and general characteristics of the zeolite are shown. In order to remove particulate matter, salts, and excess adsorbed cations, the zeolites were washed with ultrapure water until the electrical conductivity was less than 10 µS/cm (and the water appeared clear by visual inspection). The zeolites were subsequently dried at 80 °C for 24 h before beginning the experiment.

### 3.2. Column and Batch Set-Up

The experimental determination of breakthrough curves was carried out in continuous flow mode through fixed-bed columns (inner diameter d_i_: 0.9 cm; bed height H_B_: 6.2 cm; zeolite mass m_Z_: 3.6 g—Figure 5a). Before each experiment, the zeolites were contained, using a plastic mesh and glass wool, in the adsorber compartment. A bed of glass spheres (d: 0.2 mm; H_B_: 1.3 cm) was placed under the plastic mesh to achieve a uniform incoming flow to the zeolite bed. The feed was pumped (Piston pump: REGLO-CPF *Digital*; Cole-Parmer GmbH, Wertheim, Germany) in an upwards direction (flow rate $\dot{V}$: 1.84 mL/min; filter velocity v_F_: 1.75 m/h; empty bed contact time EBCT: 2.16 min). To analyze breakthrough of the sorbate in the effluent, samples were taken from 10 min to 4800 min (3.3 days).

A differential circuit batch reactor (Figure 5b) was used to investigate intraparticle diffusion (d_i_: 1.8 cm; H_B_: 3 mm; m_Z_: 0.7 g; v_F_: 29 m/h; solution volume: 1 L). A comparatively high filter velocity (29 m/h) was used to negate the impact of film diffusion. Glass spheres (d: 0.2 mm; H_B_: 9.5 cm) were placed under the plastic mesh, and glass wool was used to fix the zeolite bed in place. As for the fixed-bed column, the feed here was also pumped in an upward direction. Zeolite loading was investigated by time-dependent sampling from 30 min to 2880 min (2 days).

Batch experiments were performed using the bottle point method and were carried out in borosilicate flasks (Borosilicate 3.3 glass; VWR International GmbH; Dresden, Germany) with 100 g of zeolites per liter of water [28]. Flasks were stirred at 100 rpm (SM-30 orbital shaker; Edmund Bühler GmbH; Bodelshausen, Germany), and sampling was performed until equilibrium was reached.

The initial NH_4_^+^ and K^+^ concentrations of 0.7 mmol/L were obtained by diluting 0.3 mol/L NH_4_Cl and 0.7 mol/L KCl (analytical grade) stock solutions, respectively. To avoid microbial contamination, the water matrices used, as well as the column and batch set-ups, were autoclaved prior to experiments. Collected samples were filtered through a 0.45 µm PET filter (CHROMAFIL^®^ Xtra PET-45/25; Macherey-Nagel GmbH and Co. KG; Düren, Germany) before determining the amounts of NH_4_^+^, cations and DOC. By using an ultrapure water control, a mostly non-competitive water matrix was provided. All experiments were carried out in duplicate and at room temperature (22 °C). Before an experiment, the pH, electrical conductivity, and water temperature were recorded.

### 3.3. Breakthrough Curve Modeling

In order to reduce the mathematical effort involved in mass transfer models, the linear driving force (LDF) model was used for breakthrough curve modeling. The LDF model represents a simplified mathematical structure of intraparticle diffusion; it is widely used for the modeling of sorption processes in fixed-bed columns. Instead of Fick’s law, a mass transfer equation with LDF is used. Under the assumptions of negligible dispersion (because of high filter velocities v_F_) and validity of the Freundlich isotherm, the differential mass balance can be formulated as in Equation (1):(1)$${v_{F}}^{*}\frac{\partial c}{\partial z}+{\varepsilon_{B}}^{*}\frac{\partial c}{\partial t}+{\rho_{B}}^{*}\frac{\partial \bar{q}}{\partial t}=0$$

Equation (1) consists of three terms: (1) describes the advection with the linear v_F_ (m/h), the sorptive concentration c (mg/L), and the way z (m); (2) includes the bed porosity ε_B_ (−) and time t (h); and (3) describes the sorption for a given bed density ρ_B_ (g/cm^3^) and the average loading of the sorbent $\bar{q}$ (mg/g).

Sorption processes can be described as consisting of four sub processes: (1) advection of the sorbate within the solution near the sorbent particle; (2) film diffusion; (3) surface diffusion; and (4) pore diffusion into the sorbent particle and further sorption of the sorbate. It is generally known that the rate-limiting steps during sorption are the film and intraparticle diffusion processes.

Firstly, the film diffusion, with a linear decrease in concentration (c) to the sorbent surface, is supposedly due to a steady increase in the equilibrium concentration c_eq_ (Equation (2)). Equations (3)–(11) describe the coefficients necessary to calculate the time-dependent $\left(\bar{q}\right)$ in Equation (2).
(2)$$\frac{\partial \bar{q}}{\partial t}=\frac{k_{F}a_{v}}{\rho_{B}}\;*\;(c\;-\;c_{\mathrm{eq}})$$
(3)$$a_{v}=\frac{3}{r_{P}}\;*\;(1\;-\;\varepsilon_{B})$$
(4)$$\varepsilon_{B}=1\;-\;\frac{\rho_{B}}{\rho_{P}}$$
(5)$$k_{F}=\frac{Sh\;*\;D_{L}}{d_{P}}$$
(6)$$Sh=[2+{\left({{Sh}_{L}}^{2}+{{Sh}_{T}}^{2}\right)}^{0.5}]\;*\;[1+1.5\;*\;(1\;-\;\varepsilon_{B})]$$
(7)$${Sh}_{L}=0.644\;*\;{Re}^{\mathrm{0.5}}\;*\;{Sc}^{\mathrm{0.3}}$$
(8)ShT=0.037 * Re0.8 * Sc1+2.443 * Re−0.1 * (Sc23− 1)
(9)$$Re=\frac{v_{F}\;*\;d_{P}}{\varepsilon_{p}\;*\;\nu }$$
(10)$$Re=\frac{\nu }{D_{L}}$$
(11)$$D_{L}=\frac{3.595\;*\;10^{-14}\;*\;T}{\eta \;*\;M^{0.53}}$$

Under the simplified assumption of spherical particles with a radius r_P_ (m), the specific particle surface a_v_ of the fixed bed can be calculated. ε_B_ is a function of ρ_B_ and the apparent particle density ρ_P_ (density of water-filled particles) (g/cm^3^). The mass transport coefficient k_F_ (m/s) can be calculated by the Sherwood number *Sh* (−), the diffusion coefficient of the sorbate in water D_L_ (m^2^/s) [28], and the particle diameter d_P_ (m). There are several correlations to calculate *Sh*. In this study, the approach by Gnielinski [45] was used because the validity range of the Schmidt number *Sc* (*Sc* < 12,000) (−) was given. The Reynolds number *Re* (−) is calculated by v_F_, d_P_, ε_B_, and the kinematic viscosity ν (m^2^/s). D_L_ is a function of the temperature T (K), the dynamic viscosity η (Pa*s), and the molecular weight of the sorbent (g/mol). 

Secondly, the particle diffusion is composed of surface and pore diffusion. Under the assumption of LDF in the direction of the sorbent particle, the mass transport Equation (12) is used. Equations (13) and (14) describe the needed coefficients for the time-dependent loading q of the sorbent.
(12)$$\frac{\partial \bar{q}}{\partial t}=k_{s}^{\prime }\;*\;(q_{\mathrm{eq}}\;-\;\bar{q})$$
(13)$$k_{s}^{\prime }=k_{s}^{\prime }(0)\;*\;e^{\left(w\;*\;\bar{q}\right)}$$
(14)$$q_{\mathrm{eq},F}=K_{F}\;*\;{c_{\mathrm{eq}}}^{1/n_{F}}$$
k_s_′ (1/s) describes the effective volumetric mass transport coefficient of the surface diffusion and is often dependent on $\bar{q}$, and the empirical parameter *w* (g/mg) describes the strength of the influence of $\bar{q}$ on the intra-particle mass transfer. k_s_′(0) (1/s) is the intrinsic mass transfer coefficient. The equilibrium loading q_eq_ (mg/g) can be calculated by the Freundlich sorption model. K_F_ ($L^{n_{F}}{*\mathrm{mg}}^{1-n_{F}}$/g) characterizes the sorption strength, and n_F_ (−) determines the curve of the sorption isotherm. k_s_′(0) and *w* can be experimentally determined with the previously described differential circuit batch reactor. Furthermore, Equation (15) is used to calculate the experimental q_eq,exp_ (mg/g) of the described batch set-ups:(15)$$q_{\mathrm{eq},\exp}=\frac{c_{0}-c_{eq}}{m_{Z}}\;*\;V$$
c_0_ and c_eq_ are the initial and equilibrium concentrations (mg/L) of the sorbent in a defined volume of water V (L). m_Z_ is the sorbent mass of natural zeolites (g).

The following experimental steps and programs were used for breakthrough modeling: 

(a)Experimental determination of three different isotherms in ultrapure water with 10 pairs of values (c_eq_ and q_eq_). Isotherm 1 contains a single-sorbate system of NH_4_^+^ (c_0_(NH_4_^+^) = 0.06–2.22 mmol/L), isotherm 2 contains a single-sorbate system of K^+^ (c_0_(K^+^) = 0.03–1.02 mmol/L), and isotherm 3 contains a multi-sorbate system of NH_4_^+^ and K^+^ (c_0_(NH_4_^+^) = 0.06–2.22 mmol/L; c_0_(K^+^) = 0.7 mmol/L). The Freundlich isotherm parameters K_F_ and n_F_ were calculated with the program ISO 3.2© (Eckhard Worch 2009).(b)The pair of values (c_eq_/c_0_ and t in h) from the differential circuit batch reactor experiments and its boundary conditions (m_Z_, V, c_0_) as well as the Freundlich isotherm parameter were used to calculate the theoretical equilibrium concentration. The empirical kinetic parameters (k_s_a_v_, *w*) were adjusted manually to the course of concentration decrease by sorption. Three different kinetic experiments were conducted with the following initial concentrations in ultrapure water: (1) c_0_(NH_4_^+^) = 0.7 mmol/L; (2) c_0_(K^+^) = 0.7 mmol/L; and (3) c_0_(NH_4_^+^, K^+^) = 0.7 mmol/L. The program KIN 3.1© (Eckhard Worch 2009) determined the empirical kinetic parameters.(c)Breakthrough curves can be calculated by the program LDF 2.6© (Eckhard Worch 2009). The following boundary parameters were needed for the calculation: c_0_, K_F_, n_F_, k_F_a_v_, k_s_a_v_, m_Z_, v_F_ and ρ_B_. Furthermore, experimental data of breakthrough curves can be loaded into LDF 3.0b and compared with modeled curves. Three different experimental breakthrough curves with the same initial concentrations of step (2) in ultrapure water were recorded.

The experimental breakthrough data and the LDF model were compared to the Thomas model (Equation (16)). The Thomas model is used to describe the ion exchange of zeolites in fixed-bed columns and to estimate the maximum sorbate sorption capacity and sorption rate in the column [46].
(16)$$\frac{c_{t}}{c_{0}}=\frac{1}{1+\exp\left(\frac{k_{Th}\;*\;q_{eq}\;*\;m_{Z}}{\dot{V}}\;-\;k_{Th}\;*\;c_{0}\;*\;t\right)}$$
c_t_ (mg/L) is the sorbate concentration in the effluent at time t. The flow time t (h) is calculated as a ratio of effluent volume (L) at time t and flow rate $\dot{V}$ (L/h). k_Th_ is the Thomas model rate constant (L/(h∗mg)).

The nonlinear Thomas model fitting was performed using the program OriginPro 2023© and by simplifying Equation (16) to Equation (17) with Equations (18)–(20):(17)$$y=\frac{1}{1+\exp\left(k_{Th}\;*\;(a\;-\;b\;*\;x)\right)}$$
(18)$$y=\frac{c_{t}}{c_{0}}$$
(19)$$a=\frac{q_{\mathrm{eq}}\;*\;m_{Z}}{\dot{V}}$$
(20)$$b=c_{0}$$

### 3.4. Investigation of Natural Water Matrices

Alkali metals and alkaline earth metals are ubiquitous cations in natural water matrices. For the investigation of natural water matrices, tap water, Elbe river water, a 1:10 mix of Elbe river and tap water, and groundwater were analyzed. The NH_4_^+^ concentrations of the autoclaved water samples were measured prior to each experiment, and equal NH_4_^+^ levels were achieved by spiking with a 0.03 mol/L NH_4_Cl stock solution. To estimate the effect of cations and the DOC content on the breakthrough curves, the levels of both were measured before each experiment.

### 3.5. Analytical Methods

The standard parameters of pH, electrical conductivity, and temperature were recorded using a multimeter and sensors (Sentix^®^41 pH electrode, TetraCon^®^325 conductivity cell, Multi 340i multimeter; Xylem Analytics Germany Sales GmbH and Co. KG; Weilheim, Germany). NH_4_^+^ and cations were measured by a chromatography device (930 Compact IC; Methrom AG; Herisau, Switzerland). DOC was determined by an analyzer (TOC-VCPN Analyzer; Shimadzu Corporation, Kyoto, Japan).

## 4. Conclusions

This study is the first to apply the LDF model to the modeling of NH_4_^+^ breakthrough curves in zeolite-filled, fixed-bed columns. Investigating the influence of natural water matrices further strengthened our understanding of NH_4_^+^ breakthrough for field applications. The conclusions of our study can be summarized as follows: 

The sorption of NH_4_^+^ onto natural zeolites was more favorable than the sorption of K^+^. For this reason, natural zeolites are highly promising sorbents with which to remove NH_4_^+^ in drinking water applications.Intraparticle diffusion is the rate limiting step during NH_4_^+^ breakthrough—as determined by comparison of k_F_a_v_ and k_s_a_v_ values. The presence of K^+^ decreased the intraparticle diffusion rate of NH_4_^+^.The LDF and Thomas models were successfully validated by experimental NH_4_^+^ breakthrough data and can reduce time-consuming field investigations. The Thomas model can further reduce the complexity of experimental investigations because only breakthrough data are necessary (this is in contrast to the LDF model that is based on isothermal and kinetic data). Natural water matrices strongly decreased the treated bed volumes during NH_4_^+^ breakthrough. In particular, K^+^ and DOC concentrations have to be considered when applying natural zeolites to the treatment of drinking water.

In summary, the time-consuming and cost-intensive investigations of field experiments can be reduced by using predictions of NH_4_^+^ breakthrough when applying the LDF and Thomas models. The Thomas model is a promising tool to reduce planning and investigation costs for decentralized applications, especially in threshold and developing countries.

In future studies, we will further validate the results reported here while also studying the influence of additional parameters (e.g., competing cations, pH, organic compounds, and zeolite regeneration using brine solutions) on the breakthrough of NH_4_^+^ in zeolite-filled, fixed-bed columns.

The main findings of this study are that the LDF and Thomas models sufficiently predict the breakthrough of NH_4_^+^ in fixed-bed columns and that natural zeolites are highly promising sorbents for the removal of NH_4_^+^ in drinking water applications.

## Figures and Tables

**Figure 1 molecules-28-01614-f001:**
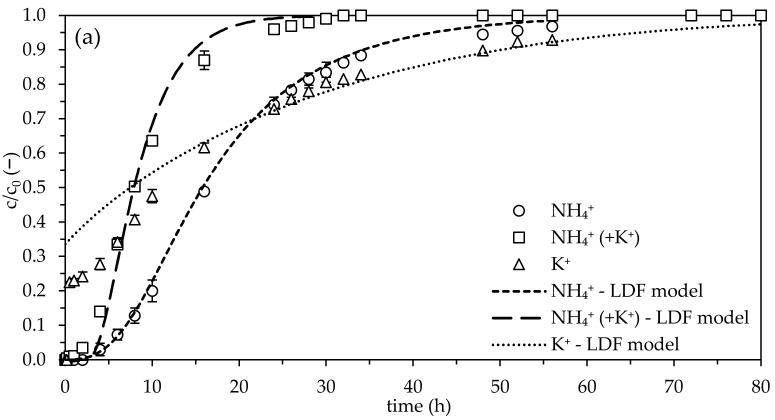
Experimental NH_4_^+^ and K^+^ breakthrough curves and their modeling using the LDF model (**a**) and the Thomas model (**b**) (matrix: ultrapure water; c_0_(NH_4_^+^) = 0.7 mmol/L, c_0_(NH_4_^+^, K^+^) = 0.7 mmol/L, c_0_(K^+^) = 0.7 mmol/L; v_F_ = 1.74 m/h; EBCT: 2.16 min; pH_0_: 5.9; T = 22 °C; *n* = 2).

**Figure 2 molecules-28-01614-f002:**
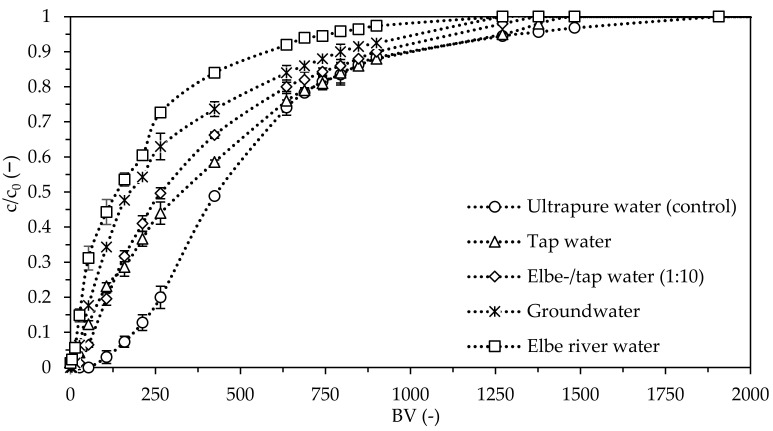
Influence of natural water matrices on the breakthrough curves of NH_4_^+^ in a natural zeolite-filled, fixed-bed column (matrix: natural; c_0_(NH_4_^+^) = 0.7 mmol/L; v_F_ = 1.74 m/h; EBCT: 2.16 min; pH_control_: 5.9; pH_0_: 7.8; T = 22 °C; *n* = 2).

**Figure 3 molecules-28-01614-f003:**
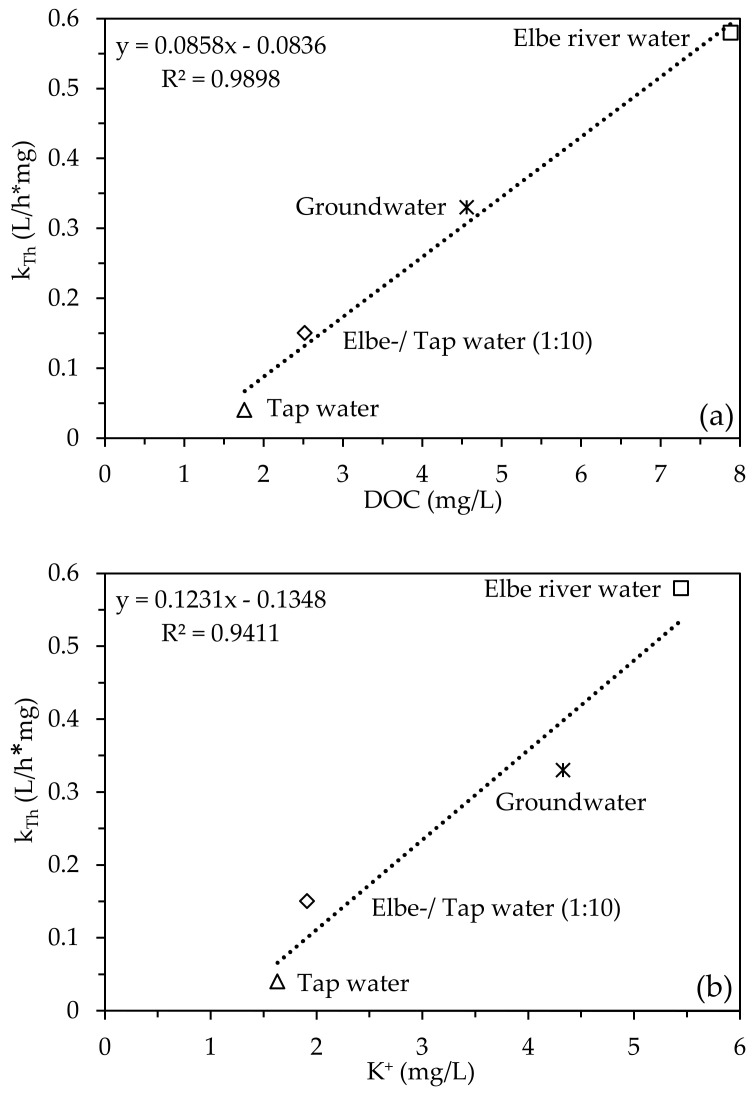
Correlation of Thomas model rate constant k_Th_ with DOC (**a**) and K^+^ (**b**) concentrations (matrix: natural; c_0_(DOC) = 1.76–7.88 mg/L; c_0_(K^+^) = 1.63–5.44 mg/L; v_F_ = 1.74 m/h; EBCT: 2.16 min; pH_0_: 7.8; T = 22 °C; *n* = 2).

**Figure 4 molecules-28-01614-f004:**
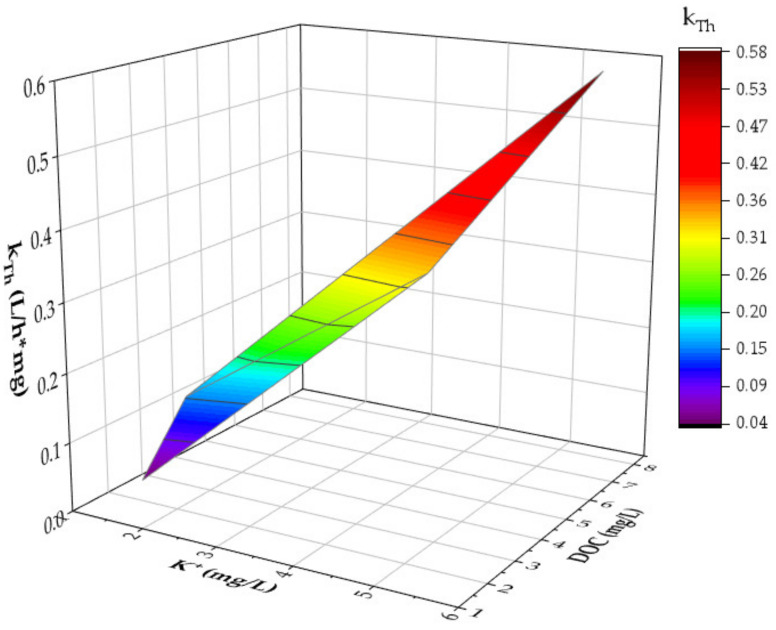
Two-factor variance analysis of k_Th_ as a function of K^+^ and DOC concentrations (matrix: natural; c_0_(DOC) = 1.76–7.88 mg/L; c_0_(K^+^) = 1.63–5.44 mg/L v_F_ = 1.74 m/h; EBCT: 2.16 min; pH_0_: 7.8; T = 22 °C; *n* = 2).

**Figure 5 molecules-28-01614-f005:**
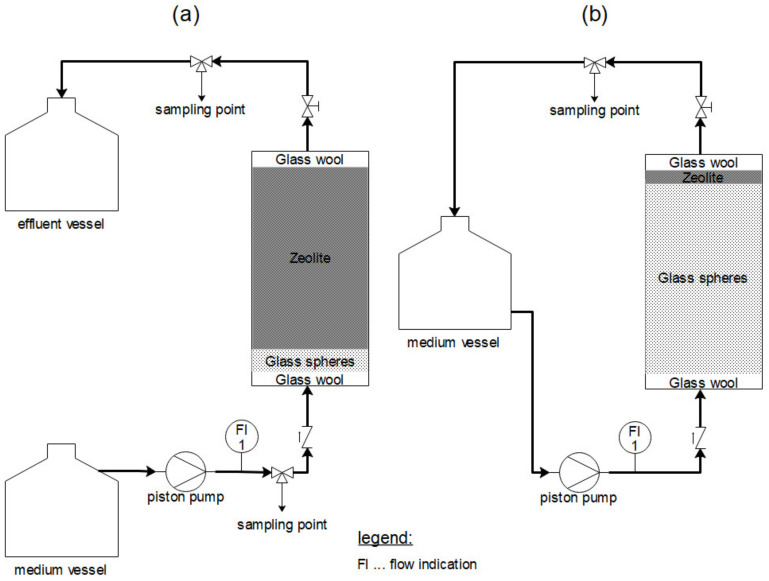
Flowcharts of the continuous flow through fixed-bed column (**a**) and differential circuit batch reactor (**b**) used in this study.

**Table 1 molecules-28-01614-t001:** Modeling parameters determined for each water matrix investigated ^x^—Parameter abbreviations can be found in the table footer ^xx^.

Investigation Steps	Modeling		Water Matrix		
	Parameter	Unit	0.7 mmol/L NH_4_^+^	0.7 mmol/L K^+^	0.7 mmol/L NH_4_^+^ (+K^+^)
Freundlich parameters	K_F_	($L^{n_{F}}{*\mathrm{mg}}^{1-n_{F}}$/g)	1.02	8.39 × 10^−3^	1.21
	n_F_	(−)	0.93	2.19	0.63
	R^2^	(−)	0.99	0.85	0.98
Diffusion parameters	k_F_a_v_	(1/s)	5.2 × 10^−1^	3.7 × 10^−1^	5.2 × 10^−1^
	k_s_a_v_	(1/s)	1.1 × 10^−4^	3.6 × 10^−5^	9.8 × 10^−5^
	*w*	(g/mg)	−0.11	−0.02	−0.05
Breakthrough curve modeling	$\dot{V}$	(m^3^/h)	1.1 × 10^−4^		
	v_F_	(m/h)	1.75		
	EBCT	(min)	2.16		
	D_L_	(m^2^/s)	2.39 × 10^−9^ (NH_4_^+^)		
			1.59 × 10^−9^ (K^+^)		
	ρ_B_	(kg/m^3^)	0.91		
	ε_B_	(−)	0.2		
	a_v_	(1/m)	7.61 × 10^3^		
	ν	(m^2^/s)	9.55 × 10^−7^		
	Re	(−)	1.64		
	Sc	(−)	3.98 × 10^3^		

^x^ Mean values were used for modeling because of <5% standard deviations in all experiments. ^xx^ Abbreviations: K_F_, n_F_: Freundlich isotherm parameters; k_F_a_v_: mass rate constant of the film diffusion; k_s_a_v_: mass rate constant of the intraparticle diffusion; *w*: describes strength of the average loading of the sorbent on the intraparticle mass transfer; $\dot{V}$: volume rate; v_F_: filter velocity; EBCT: empty bed contact time; D_L_: diffusion coefficient in the aqueous phase; ρ_B_: bed density; ε_B_: bed porosity; a_v_: specific particle surface; ν: kinematic viscosity; Re: Reynolds number; Sc: Schmidt number.

**Table 2 molecules-28-01614-t002:** Chemical composition and zeolite characteristics, data provided by supplier.

Composition	Value (%)	Characteristics	
SiO_2_	65.00–71.30	Exchange capacity	1.2–1.5 mol/kg
Al_2_O_3_	11.50–13.10	Selectivity	NH_4_^+^ > K^+^ > Na^+^ > Ca^2+^ > Mg^2+^
CaO	2.70–5.20	Mean pore diameter	0.4 nm
K_2_O	2.20–3.40	Specific surface	30–60 m^2^/g
Fe_2_O_3_	0.70–1.90	Si/Al	4.80–5.40 (−)
MgO	0.60–1.20	Grain sizes	0.5–0.8 mm
Na_2_O	0.20–1.30		
TiO_2_	0.10–0.30		

**Table 3 molecules-28-01614-t003:** Initial cation and DOC concentrations of the natural water matrices investigated.

Water Matrices	Cations (mmol/L)				DOC (mg/L)
	K^+^	Na^+^	Mg^2+^	Ca^2+^	
Ultrapure water (control)	<CR ^x^	<CR ^x^	<CR ^x^	<CR ^x^	0.09
Tap water	0.04	0.36	0.12	0.83	1.76
Elbe-/tap water (1:10)	0.05	0.43	0.13	0.84	2.52
Groundwater	0.11	0.91	0.66	1.23	4.56
Elbe river water	0.14	1.16	0.31	0.92	7.88

^X^ below calibration range: 0.2–40 mg/L.

**Table 4 molecules-28-01614-t004:** Channel dimensions and ring types of natural zeolites (clinoptilolite ^x^).

Channel Dimension (nm)		Ring Type	
Inglezakis and Zorpas [39]	Margeta et al. [40]	Number	Pore Structure
0.75 × 0.31	0.72 × 0.44	10	2-dimensional
0.47 × 0.28	0.55 × 0.40	8	
0.46 × 0.36	0.47 × 0.41	8	

^x^ Framework type HEU—cell parameters: a = 1.75 nm, b = 1.76 nm, c = 0.74 nm [41].

**Table 5 molecules-28-01614-t005:** Physical characteristics of the investigated cations.

Cation	Charge Density (C/mm^3^)	Ionic Radius (nm)	Hydrated Radius (nm)
NH_4_^+^	12	0.148 ^(a)^	0.331 ^(c)^
K^+^	14	0.138 ^(b)^	0.331 ^(c)^
Na^+^	37	0.102 ^(b)^	0.358 ^(c)^
Ca^2+^	76	0.100 ^(b)^	0.412 ^(c)^
Mg^2+^	205	0.072 ^(b)^	0.428 ^(c)^

^(a)^ [42],^(b)^ [43],^(c)^ [44].

## Data Availability

Data are available upon request.

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
