# Peer review of "Natural Zeolites for the Sorption of Ammonium: Breakthrough Curve Evaluation and Modeling"

_molecules, 2023, doi:10.3390/molecules28041614_

Round 1
Reviewer 1 Report
In general, the work of Eberle et al. is well presented and written. However, there are some issues that must be addressed before it is suitable for publication:
1. Since the title and focus of the work is modeling the breakthrough curve, why did not you try to model the data of different water samples (Figure 2)?
2. The fitting using any of the models proposed in your work is not perfect, so that I do not see how you can conclude this: “…can reduce time-consuming field investigations” in line 486.
To withdraw such a conclusion, you may need to analyze much more experimental data and find a model that is able to describe as much experimental data as possible.
Author Response
Please find our response in the attached PDF file.

Author Response

(The authors gave the same response as above.)

Reviewer 3 Report
This work reports on a modeling analysis of adsorption behavior of ammonium ion onto natural zeolites for water purification. The authors obtained the NH4+ and K+ breakthrough curves experimentally and analyzed these data using the LDF model and the Thomas model.
Those results showed that the LDF and Thomas model could predict the breakthrough behavior of ammonium ion in fixed-bed columns for degradation ammonium ion in water.
The results seem to be useful not only for water purification using natural zeolites, but also for adsorption of by-products using fixed-bed columns in flow reactors for organic synthesis.
I think this work is acceptable after minor revision.
My comment shown below.
1. P. 3, Table 1.
The bed density value is too small. The value of 0.91 kg/m3 is closer to that of air than water. Please check it.
2. P.8, Table 3.
It is desirable to add the mineral names and framework type codes of the natural zeolites shown in the Table.
Author Response

(The authors gave the same response as above.)
